# Usefulness of Contrast-Enhanced Ultrasound in the Differentiation between Hepatocellular Carcinoma and Benign Liver Lesions

**DOI:** 10.3390/diagnostics13122025

**Published:** 2023-06-10

**Authors:** Adam Dobek, Mateusz Kobierecki, Wojciech Ciesielski, Oliwia Grząsiak, Adam Fabisiak, Ludomir Stefańczyk

**Affiliations:** 1Department of Radiology and Diagnostic Imaging, Norbert Barlicki Memorial Teaching Hospital No. 1, Medical University of Lodz, 90-153 Lodz, Poland; 2Department of General Surgery and Transplantology, Norbert Barlicki Memorial Teaching Hospital No. 1, Medical University of Lodz, 90-153 Lodz, Poland; 3Department of Digestive Tract Diseases, Norbert Barlicki Memorial Teaching Hospital No. 1, Medical University of Lodz, 90-153 Lodz, Poland

**Keywords:** CEUS, magnetic resonance imaging, computed tomography, ultrasonography, contrast, hepatocellular carcinoma, hepatocellular adenoma, focal nodular hyperplasia

## Abstract

A differentiation between hepatocellular carcinoma (HCC) and benign liver lesions is required. The aim of the study was to perform an analysis of the time of enhancement of focal liver lesions in a contrast-enhanced ultrasound (CEUS) examination. The curves of enhancement and the homogeneity of the tumor enhancement were assessed. The study included 52 patients with diagnoses of hepatocellular adenoma (18), focal nodular hyperplasia (11) and HCC (28). The study included magnetic resonance imaging or computed tomography and a comparison of the obtained information with CEUS. In the benign lesions groups after 20–30 s, the enhancement was similar to the liver parenchyma. In the HCC group, the enhancement was slightly less intense compared to the liver parenchyma and the benign lesions. The difference of the enhancement in the arterial phase (benign lesions vs. HCC) was *p* = 0.0452, and the difference of enhancement in the late venous phase (benign lesions vs. HCC) was *p* = 0.000003. The homogeneity of the enhancement (benign lesions vs. HCC), respectively, was *p* = 0.001 in the arterial phase, *p* = 0.0003 in the portal venous phase and *p* = 0.00000007 in the late venous phase. Liver tumors can be classified as benign when they are homogenous in the arterial phase and don’t present washout. HCC in the arterial phase is inhomogeneous and washout is observed in the venous phases. When radiological symptoms suggest malignant lesion, CEUS can be used to select the best biopsy access.

## 1. Introduction

Hepatocellular carcinoma (HCC) is the sixth most common tumor worldwide and the most common malignant primary tumor of the liver. Its occurrence is above 80% and the frequency is constantly rising [1,2,3]. HCC usually develops based on cirrhosis, hepatitis B/C or on account of the progression of hepatocellular adenoma (HCA) to HCC [1,4,5]. Thus, a differentiation between malignant and benign liver lesions is required. HCA and focal nodular hyperplasia (FNH) should be especially mentioned due to similar patterns of enhancement and their potentials for growth. FNH can cause compression only on the surrounding tissues, whereas HCA can succumb to divulsion, hemorrhage or, as mentioned above, progression to HCC. The correlation of the HCA occurrence in the population of young women using oral contraceptives is airtight [5,6,7]. The correlation between oral contraceptives and the occurrence of FNH is being discussed [8]. Both tumors behaved similarly in a CEUS examination where Roche et al. denoted HCA as “FNH mimickers” [9]. The “wash-in” effect was observed approx. 20 s after the contrast administration. After its enhancement aligned, the lesions remained iso or slightly hypo-enhanced compared to the liver parenchyma in the venous phase and late venous phase of the examination. Due to the early differentiation and regular observation of both tumors, it was possible to notice the changes in the enhancement and instigate the treatment in case of the HCA progression to HCC. In current guidelines, biopsy is recommended as a gold standard in focal liver lesion diagnosis [10]. However, it has some limitations. The acquisition can be missed, and there is a risk of bleeding and a potential spreading of the tumor on the surrounding tissues [11,12]. Furthermore, there is a possibility that a sample may contain only hepatic cells, which in case of the above-mentioned tumors, is not diagnostic. The key features that allow for the recognition of HCC are the presence of the “washout” effect and the inhomogeneous enhancement of the lesion. Washout is defined as a reduction in the enhancement in part or in the whole lesion compared to the liver parenchyma. Washout in HCC is usually mild and characterized by a late onset (>60 s, usually >90 s) [1,3,12,13]. The aim of the study was to perform an analysis of the time of enhancement of focal liver lesions using a CEUS examination. Furthermore, the curves of enhancement and the homogeneity of the tumor enhancement were assessed.

## 2. Materials and Methods

The study was conducted according to the guidelines of the Declaration of Helsinki and was approved by the Bioethics Committee at the Medical University of Łódź. All the patients gave their informed consent to participate in the project. The criteria of inclusion was the confirmation of focal liver lesions using magnetic resonance imaging (MR)/computed tomography (CT) examination. The CEUS was performed up to 48 h after MR/CT. The criteria of exclusion, according to recommendation of SonoVue producer, was respiratory insufficiency, acute coronary syndrome, adverse post-contrast reactions, or declaration of pregnancy.

### 2.1. Patients

A retrospective study was performed in the Department of Radiology of USK 1 im. Norberta Barlickiego at the Medical University of Łódź. The study included 52 patients with clinical diagnoses of hepatocellular adenoma (18 tumors), focal nodular hyperplasia (11 tumors) and hepatocellular carcinoma (28 tumors). Two lesions of the FNH type were observed in one patient. Three lesions of the HCA type were observed in one patient (in this case, a CEUS examination was performed twice and every lesion was considered as a separate observation). The study included MR or CT, and the obtained information was compared with the contrast-enhanced ultrasound. Both procedures were performed with a time interval of less than 48 h. The 1.5 and 3 T scanners (Siemens Magnetom Avanto, Siemens Magnetom Vida), GE Healthcare Revolution CT and GE Lightspeed VCT 64 Slice were used. The MR and CT examinations were performed according to the LI-RADS Version 2018 [14], taking into account the post-contrast sequences with the assessment of late enhancement (0.1 mL of Gadovist 1.0 per kg body weight—MR; 1 mL of Ultravist-370 per kg body weight—CT). Every benign liver lesion diagnosis was confirmed using MR. In addition, core needle biopsy was performed in two cases and CT was performed in five cases. The diagnosis of every HCC was performed using CT and confirmed using core needle biopsy sampling. In addition, MR was conducted in eight patients (Table 1). The size of the lesions was assessed in the early post-contrast phase. The differentiation between HCA and FNH was based on the presence of a central fibrous scar, whereas the differentiation of the mentioned lesions from HCC was based on the presence of inhomogeneous enhancement and the wash-out effect.

### 2.2. Imaging Technique

Before the CEUS, the MRI/CT examination in the selected protocol with contrast was performed for the confirmation of the focal liver lesion diagnosis. The CEUS was performed according to the guidelines for the CEUS in liver—2020 update [15]. The GE logiq 7 system with the convex probe 4C was used. The examination included a standard grey scale (B-mode) ultrasound examination of the liver. This part of the examination was performed to evaluate the presence of potential lesions which did not exhibit a contrast enhancement and could be misdiagnosed as malignant infiltration. The size, localization and quantity of the lesions were recorded. Next, color Doppler imaging was performed. In the last step, the CEUS was performed starting with an injection of 2.4 mL of a contrast agent (SonoVue) in a medial cubital vein. The CEUS was c conducted using a low mechanical index (<0.1) to avoid the destruction of the contrast agent bubbles [1,9]. Three main phases of acquisition were performed. The acquisition of each of the phases was achieved in the following intervals: the arterial phase (10–45 s), the portal venous phase (45–120 s), the late venous phase (120–640 s). The acquisition was performed up to 640 s. However, in some cases, the data ended after 180 s of examination. The dynamic enhancement profile change wasn’t noticeable after 120 s of examination. Therefore, we decided to present our analysis in 0–180 s range. During the examination, the enhancement of the tumor was compared to the liver parenchyma (Figure 1).

### 2.3. Statistical Analysis

A statistical analysis of the demographic data and size of the lesions was performed. The differences between the patient characteristics were assessed using a Mann–Whitney U test for the categorical data. For the comparison of the dichotomous variables, the chi-square test with Yates correction was used. Next, the case-by-case HCA, FNH and HCC enhancements in the CEUS were analyzed in 10 s intervals for a total observation time of 180 s. If a case had a single missing observation, a linear interpolation was used to fill in the gaps. Missing observations were caused by respiratory movement during the acquisition. The mean enhancement values for each tumor over time were calculated. A single-case HCA, FNH and HCC enhancement was also presented. The unchanged liver parenchyma enhancement was assessed and compared with the tumor-changed organ. The circular region of interest (ROI) of the whole tumor and the best and the worst enhanced parts of the tumor were marked, covering approximately half of the tumor (Figure 2A–C). The acquisition of the echogenicity of the marked regions was measured in every phase of the examination—arterial phase, portal venous phase and late venous phase. The distributional characteristics of the difference between the best and the worst part of the tumor is presented on a box plot. The median is marked by the orange line in the center of the box. No outliers were present in the collected data. All the graphs were created using Python matplotlib v3.6.2. As a result of the small sample size, there was not enough statistical power to test the normality of the data. Due to similar diagnostic algorithms in the case of benign lesions, we compared the group of benign lesions (HCA + FNH) to the HCC group. Box plots and the area under the receiver operator characteristic curves (ROC) were used to assess the differences between the groups. The differences between the echogenicity of the benign lesion enhancement compared to the HCC enhancement were assessed using a non-parametric Mann–Whitney U test. The difference was tested in the arterial phase (20 s) and the late venous phase (120 s). In 120 s, few observations provided no data and the tested group was smaller (23 observations and 18 observations, respectively). The same test was used to assess the differences between the homogeneity of the tumor enhancement in each phase. A *p*-value less than 0.05 was marked as statistically significant.

## 3. Results

The estimated size of the lesions based on the post-contrast sequences in the MR/CT examinations ranged from 0.64 cm to 8.6 cm (mean 3.25 cm) in the adenomas group, from 1.4 cm to 4.6 cm (mean 3.2 cm) in the FNH group and from 2.8 cm to 18.7 cm (mean 7.33 cm) in the HCC group. The estimated size of the lesions based on the post-contrast sequences in the CEUS studies ranged from 0.66 cm to 8.2 cm (mean 3.31 cm) in the adenomas group, from 1.41 cm to 4.6 cm (mean 3.2 cm) in the FNH group and from 2.88 cm to 18.63 cm (mean 7.22 cm) in the HCC group (Table 2 and Appendix A). The performed statistical testing of the demographic data and size of the lesions demonstrated that the data within the benign lesion groups was similar. The ROC curves also presented that the echogenicity of the HCA and FNH groups during the examination was similar. According to the obtained information, the benign lesions were compared to the HCC group (Figure 3). In adenomas and the FNH group, a similar tumor enhancement profile was observed compared to the remaining liver parenchyma (Figure 4, Figure 5 and Figure 6). In the HCC group, the washout effect and hypo-enhancement compared to the remaining liver parenchyma in the later phases of examination was observed. In adenomas and the FNH group after 20–30 s, the enhancement of the focal lesion was similar to the liver parenchyma. In the HCC group, the enhancement was slightly less intense compared to the liver parenchyma and benign lesions. The mean values of enhancement for every group are presented in Figure 7. In adenomas and the FNH groups, the values remained comparable until the end of the observation (tumor background gain difference was ±14 dB). In the HCC group, the values of the tumor background difference changed from ±10 dB in the arterial phase to ±19 dB in the late venous phase. The difference of enhancement in the arterial phase (28 observations in the HCA + FNH group vs. 28 observations in the HCC group) was *p* = 0.0452, and the difference of enhancement in the late venous phase (23 observations in HCA + FNH group vs. 28 observations in HCC group) was *p* = 0.000003. The difference was caused due to the washout effect in HCC (Figure 8). The ROC curves presented the difference between the echogenicity of the benign lesions group vs. the HCC group during the examination. The specificity and sensitivity of the CEUS in the differentiation between the benign lesions and HCC is presented (Figure 9). In the HCC subgroups divided for lesions <4 cm and lesions >4 cm, including all the observations, the difference of homogeneity was visible. The difference of the tumor echogenicity between the selected smallest lesions and all the group of HCC could be up to ±25 dB (Figure 10). The difference between the best and the worst enhanced part of the tumors did not exceed 4 dB in the FNH group and 3 dB in the HCA group. In HCC, the difference between the best and the worst enhanced part of the tumor did not change significantly in the arterial (9 dB), early venous (7 dB) and late venous phases (5 dB) (Figure 11). The homogeneity of enhancement (28 observations HCA + FNH group vs. 28 observations in HCC group), was *p* = 0.001 in the arterial phase, *p* = 0.0003 in the portal venous phase and *p* = 0.0000007 in the late venous phase.

## 4. Discussion

HCA and FNH occur frequently in the younger population, and thus require long term observation to detect potential symptoms of transformation into HCC. The important risk factors for HCC are cirrhosis, hepatitis B/C and alcoholism [3,4]. HCC in CT and MR enhances the intensity in the arterial phase compared to the liver parenchyma and presents a washout effect during the portal venous or the delayed phases [16,17]. Furthermore, HCA can undergo a progression to HCC. A typical HCC image in a CEUS is characterized by a mild washout with a late onset (>60 s) and a strong enhancement in the arterial phase. The enhancement of a tumor is often inhomogeneous due to the areas of potential necrosis or the presence of a hemorrhage. The washout in HCC can start within 2–3 min of an examination. Therefore, an important feature for examination in case of HCC suspicion is to provide a late venous phase acquisition [1,3,11,12,13,18]. Schellhaas et al. and Strobel et al. acknowledged that an uncharacteristic behavior of HCC in a CEUS examination should be considered during examination with an unspecific course [3,18]. HCA and FNH are characterized by early, homogenous enhancement in the arterial phase and persistent iso-enhancement in the venous phases compared to the liver parenchyma. Our results suggest that there were three major features that allowed for the differentiation between benign lesions and HCC. First, the values of enhancement in the arterial phase were slightly weaker in case of HCC compared to the benign lesions (around 6 dB). Second, the washout effect in the late phase of the examination was observed. The difference in enhancement between the benign lesions and HCC was in the range of 10–15 dB. Finally, there was a big difference in the homogeneity of the analyzed tumors in all the phases of examination. If a tumor was big and presented all of these radiological features, HCC could be diagnosed with significant possibility. Some authors reported that in the case of small HCC (1–2 cm), the enhancement in the arterial phase did not have to achieve inhomogeneity or present the washout criteria [11,12,19]. Our observations did not include lesions that could be qualified as this group. However, we observed, especially in HCC <4 cm, that their homogeneity was higher compared to the lesions >4 cm and all of the groups. The group of HCC <4 cm in our study was too small to state clearly. However, the tendency suggested that if a lesion was smaller, the enhancement was more homogenous, which can be misleading in the differentiation of the benign lesions. In cirrhotic patients, the iso-enhancement in the late phases of examination did not exclude the presence of a malignant lesion [9,13]. The ancillary features that can help distinguish malignant from benign lesions included a size measurement of >2 years. An increase indicated a malignant lesion, whereas a size or stability reduction indicated a benign tumor [12]. Golfieri et al. emphasized that, especially in case of small lesions, diagnostic alertness is crucial. The early detection of HCC makes the disease potentially curable [16]. Therefore, periodic surveillance is desirable. Renzulli et al. reported that there are no generally accepted strategies for the management of nodules that don’t present typical radiological symptoms of HCC in cirrhotic liver [17]. Bartolotta et al. suggested performing a control CEUS every 6 months [13]. Today, CEUSs have obtained increasingly wider acceptance in the diagnostics of focal liver lesions. The accuracy of a CEUS compared to MRI and CT is similar. Furthermore, this method is cheaper, faster and more available for patients. In addition, a CEUS doesn’t require a large dose of a contrast agent, unlike CT, so it can be used in patients suffering from renal failure, renal obstruction or allergies [2,4,13]. Despite the multiple advantages of CEUS, it has some limitations. Patient body habits, the presence of bowel gas and the size and localization with complicated reachability within the liver can handicap the diagnosis. Only the imaging of part of the liver is possible at a given moment [4]. Today, CEUSs are the object of discussion as to whether they should be included as recommendations of focal liver lesion diagnosis. Despite the development of imaging, biopsy and histopathological verification are regarded as the gold standard for focal liver lesion diagnosis. However, these methods, both radiological and histological, have some shortcomings. It should be noted that biopsy is an invasive procedure, which can be potentially harmful for patients. It can trigger a tumor seeding or hemorrhage, especially in the case of cirrhotic patients [11]. The weakness of histological diagnosis in the case of liver tumors, including HCA, FNH and HCC, is their histological similarity since each of them is built from hepatic cells. Therefore, an aspirated sample can contain only hepatic cells and doesn’t have to be diagnostic. Nevertheless, biopsy should be considered in every case as a part of the diagnostic path. However, it should be considered that in some cases carrying out this procedure will not be possible or can cause serious complications. Today, the need for biopsy is widely discussed. Schellahaas et al. and Terzi et al. suggested that biopsy should be performed due to the possibility of overlooking HCC in cirrhotic liver in radiological imaging. They indicated that it is better to treat a dysplastic nodule as HCC, because less harm is done to the patient than if HCC is overlooked [3,11]. Geyer et al. indicated that HCC, unlike many other solid malignant tumors, can be successfully diagnosed only by imaging without tissue sampling [2]. However, Schellahaas et al. challenged this sentence due to the possibility of atypical HCC behavior in CEUSs [3]. Nevertheless, if there is a decision to perform a biopsy, a CEUS can be used as a great tool to indicate the best place to puncture. The latest studies confirmed the thesis that CEUS-guided biopsy has a higher diagnostic accuracy than US-guided biopsy, especially in small lesions <2 cm and in HCC [14,20,21]. The fact that CEUS increases the diagnostic accuracy in these two groups is promising. Our observations in the HCC group <4 cm seemed to confirm that small HCC are more homogenous than larger lesions (>4 cm). Therefore, when there are suspicions about the character of a small lesion with homogenous enhancement, CEUS-guided biopsy should be considered. Taking into account these problems, we would like to highlight that HCC diagnostics require a good and organized protocol. There must be a strong suspicion of HCC to deploy a full diagnostic tool. The primary diagnosis of HCC should be performed using CT. In the case of a malignant lesion suspicion, a late venous phase acquisition is required. However, it should be highlighted that performing CT in every patient with lesions within the liver exposes them to unnecessary radiation and generates logistic problems and costs. Furthermore, the potential limitations of biopsy should be remembered. HCC should be diagnosed as fast as possible to perform effective treatment. The diagnostic algorithm is constantly submitted for analysis to upgrade it. Today, the role of gadoxetic acid-enhanced MRI is emphasized. It allows for diagnosing high-grade dysplastic nodules and HCC in the early stage in cirrhotic liver. It was reported that gadoxetic acid-enhanced MRI has a higher sensitivity and specificity for correct HCC diagnoses than CT and MRI using other contrast agents [16,17]. In light of the need for diagnostic algorithm improvement, we postulate that CEUSs can be considered a great tool for the differentiation of HCC from benign and other malignant focal liver lesions. In the case of detecting malignancy symptoms in a CEUS, there is indication to perform extended diagnostics. During further examination, a CEUS can help designate the best access for biopsy. The limitations of this study include the difference in the sizes of the benign lesions compared to HCC. The HCC lesions were usually bigger than HCA and FNH. Furthermore, the remaining liver parenchyma of the patients who suffered from HCC was often pathological, which handicapped the comparison to the focal lesion. Collecting larger groups of patients will allow us to verify our observations and perform deeper statistical analyses. Today, EASL allows for diagnosing benign focal liver lesions only with use of diagnostic imaging. Biopsy is suggested only in doubtful cases [22]. Therefore, the majority of benign lesions in our study did not have a histopathological confirmation of the diagnosis. Furthermore, CEUS examinations can be performed only on one plane, so a three-dimensional structure was impossible to obtain. Additionally, the presence of a CEUS in regular use is short and the efficiency of the proposed algorithm must be confirmed in a longer follow up.

## 5. Conclusions

Liver tumors can be classified as benign when they are homogenous in the arterial phase and do not present a washout effect. The presence of a central fibrous scar allows for recognizing FNH and distinguishing it from other benign lesions, such as HCA. Malignant hepatic lesions, such as HCC, in the arterial phase present an inhomogeneous enhancement pattern due to the presence of bleeding or necrosis. These areas within a tumor should be avoided during biopsy due to lack of the diagnostic value of the sample. In the venous phases, a washout is observed. In the case of a lack of washout, a benign lesion should be considered. When the radiological symptoms suggest a malignant lesion, CEUS can be used to decide whether biopsy should be conducted and to select the best biopsy access. These features can be observed in CEUS as well as in CT or MR. Therefore, we suggest that CEUS can be used for selecting patients for CT/MR examination and providing the optimal placement of biopsy by avoiding areas of necrosis or bleeding. Additionally, we believe CUES can improve the sensibility and sensitivity of diagnoses. In consequence, it should improve the percentage of correct diagnoses.

## Figures and Tables

**Figure 1 diagnostics-13-02025-f001:**
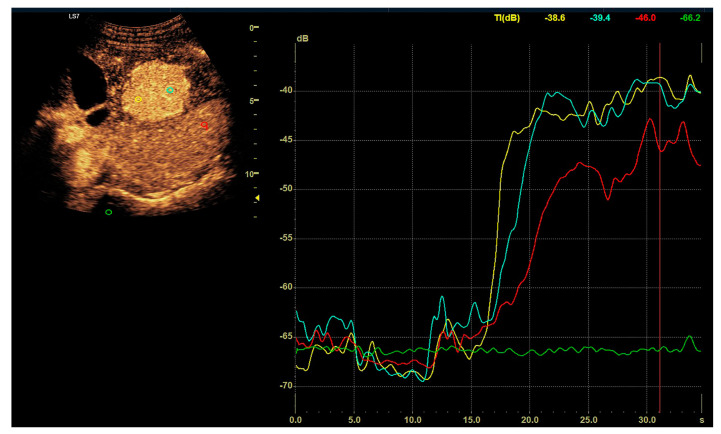
Assessment of the degrees of enhancement in the post-contrast examination (CEUS). The regions of interest (ROI) were placed within the lesion (two areas: blue and yellow). Another area was placed in the parenchyma of the liver (red). Widening of scale of graph (green). The enhancement curves were recorded for 2–3 min in sequences of approx. 20–30 s. The values burdened with motor artifacts causing the dislocation of the areas of interest were eliminated.

**Figure 2 diagnostics-13-02025-f002:**
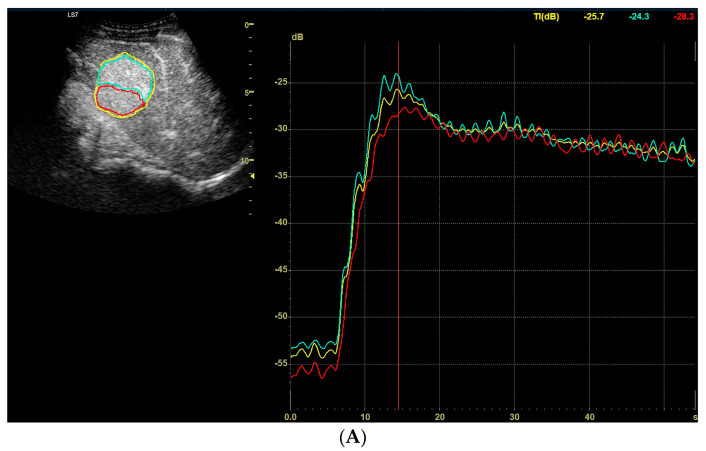
(**A**–**C**) Assessment of the degrees of enhancement in the post-contrast examination (CEUS). Areas of interest embraced the whole tumor (yellow), the best (blue) and the worst (red) enhanced part of lesion. The enhancement of the curves were recorded three times—in the arterial phase, portal venous phase and late venous phase, respectively, for hepatocellular adenoma, focal nodular hyperplasia, hepatocellular carcinoma.

**Figure 3 diagnostics-13-02025-f003:**
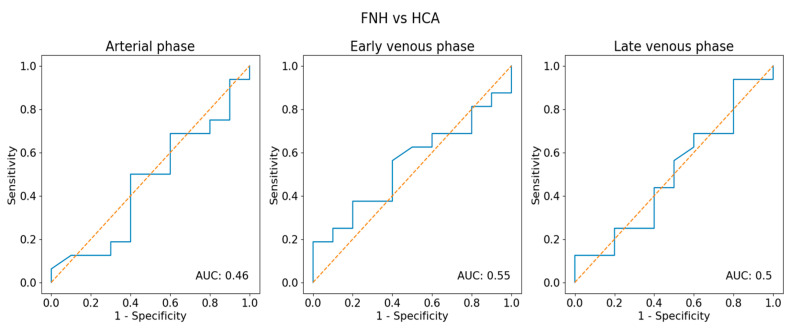
Receiver operator characteristic (ROC) curves analysis for predicting the focal nodular hyperplasia (FNH) group or hepatocellular adenoma (HCA) group based on the echogenicity of a tumor. AUC—area under the curve of the ROC.

**Figure 4 diagnostics-13-02025-f004:**
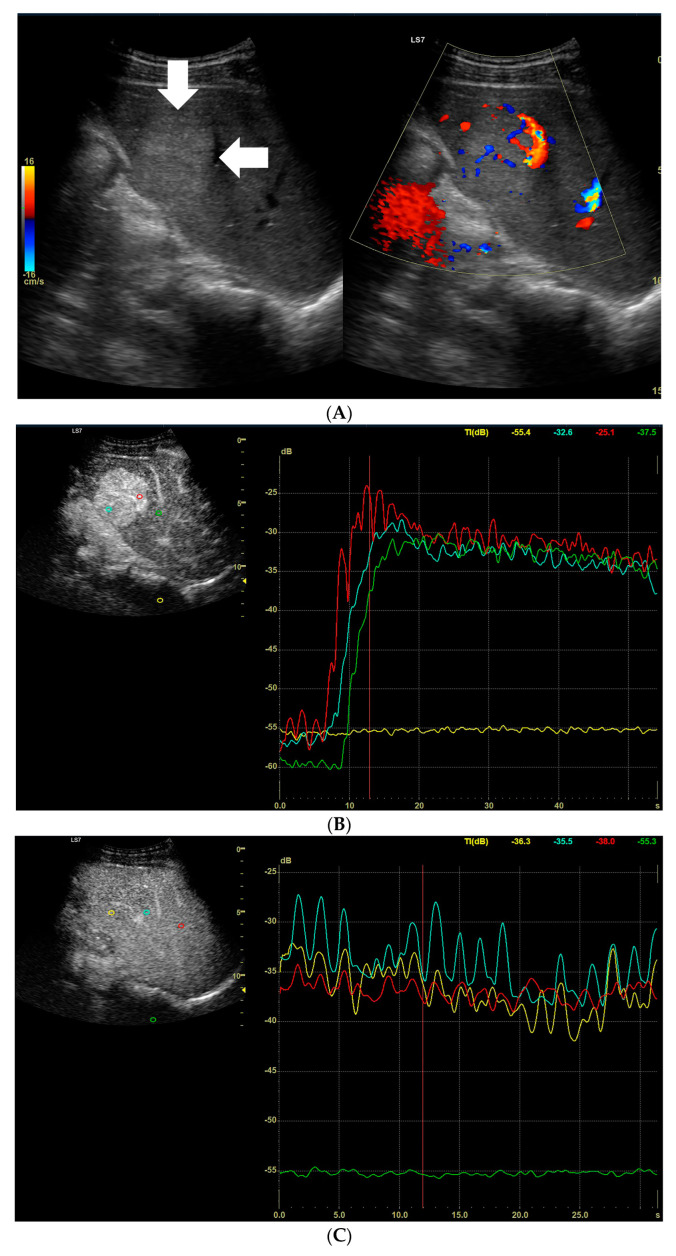
(**A**) Image of an ultrasound and color doppler in the transverse plane, focal lesion in the right lobe of the liver identified using MRI as HCA (full white arrows). The echogenicity of the lesion was slightly higher in relation to the liver parenchyma. (**B**) Image of the CEUS. The curves of enhancement in the early phase of examination. The tumor area-red and blue lines. Slightly less intensive and delayed enhancements the liver parenchyma-green line. The disturbance of the curves in the early phase of the study (0–5 s) was caused by the displacement of the liver due to breathing. (**C**) Image of the CEUS. The curves of enhancement in the areas of interest in the venous phase of the study are aligned. The tumor area-blue and yellow line. Slightly less intensive enhancement of the liver parenchyma-red line. (**D**) Image of the CEUS. The curves of enhancement in the areas of interest in the late venous phase of the study are aligned. The tumor area-blue and yellow line. Enhancement of the liver parenchyma is similar to tumor-red line.

**Figure 5 diagnostics-13-02025-f005:**
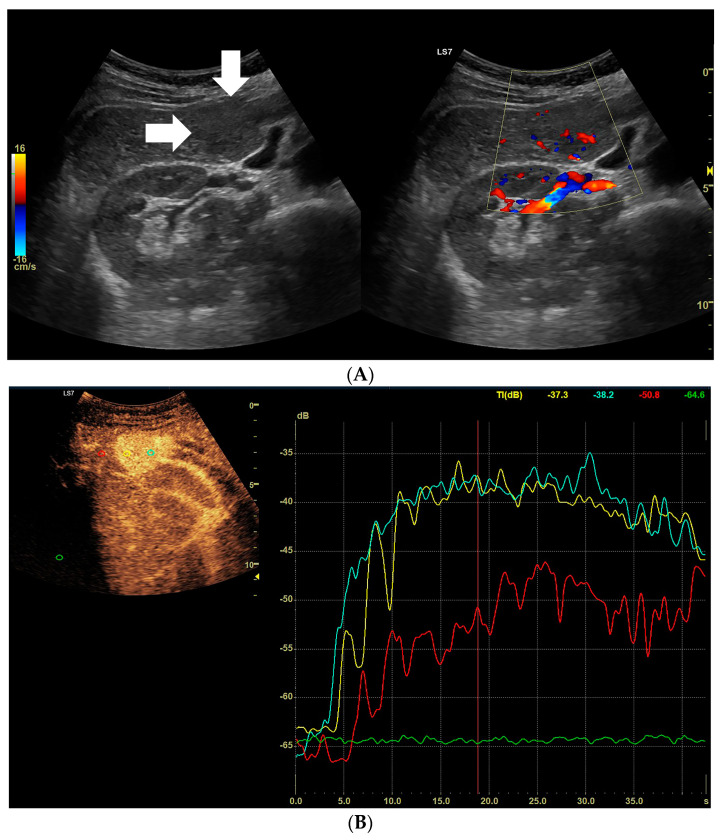
(**A**) Image of an ultrasound and color doppler in the transverse plane, focal lesion in the right lobe of the liver identified using MRI as FNH (full white arrows). The echogenicity of the lesion was slightly higher in relation to the liver parenchyma. In the central part, a small scar is better visible in the color doppler technique. (**B**) Image of the CEUS. The enhancement curves in the early phase of examination. The tumor area-yellow and blue lines. Less intensive and delayed enhancement of the liver parenchyma-red line. (**C**) Image of the CEUS. The curves of enhancement in the venous phase of examination are aligned. Area of tumor-blue and red lines. Enhancement of the liver parenchyma similar to the tumor enhancement-green line. (**D**) Image of the CEUS. The curves of enhancement in the late venous phase of examination are aligned. Area of tumor-blue and red lines. Slightly less intensive enhancement of the liver parenchyma-red line.

**Figure 6 diagnostics-13-02025-f006:**
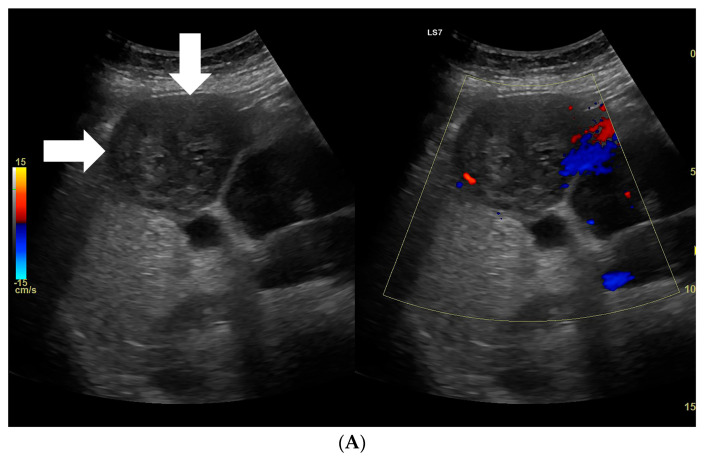
(**A**) Image of an ultrasound and color doppler in the transverse plane, focal lesion in the right lobe of the liver identified using CT as HCC (full white arrows). The lesion was hypoechogenic in relation to the liver parenchyma. Within the lesion, multiple necrosis areas are visible. (**B**) Image of the CEUS. The enhancement curves in the early phase of examination. The tumor area-red and blue lines (necrosis areas). Stronger and earlier enhancement of the liver parenchyma-yellow line. (**C**) Image of the CEUS. The curves of enhancement in the venous phase of examination. Area of tumor-blue and yellow lines (necrosis areas). Enhancement of the liver parenchyma stronger than the tumor enhancement-red line. The washout effect is visible. (**D**) Image of the CEUS. The curves of enhancement in the late venous phase of examination. Area of tumor-red and blue lines (necrosis areas). Stronger enhancement of the liver parenchyma-yellow line. The lesion is hypo-enhanced compared to the liver parenchyma; the washout effect continues. Widening of scale of graph (green).

**Figure 7 diagnostics-13-02025-f007:**
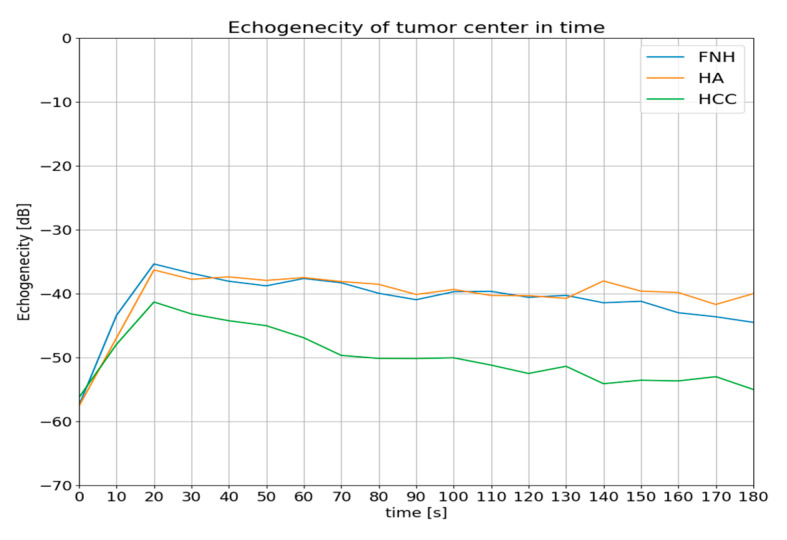
Mean values of the difference in the tumor enhancement and the remaining liver parenchyma over time recorded in the hepatocellular adenoma (HCA) group (blue line), in the focal nodular hyperplasia (FNH) group (orange line) and in the hepatocellular carcinoma (HCC) group (green line).

**Figure 8 diagnostics-13-02025-f008:**
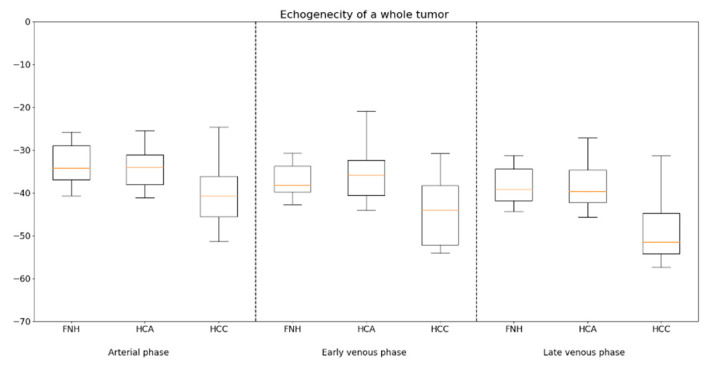
Record of the values of enhancement of the whole tumor recorded in the arterial, portal venous and late venous phases. focal nodular hyperplasia (FNH), hepatocellular adenoma (HCA), hepatocellular carcinoma (HCC).

**Figure 9 diagnostics-13-02025-f009:**
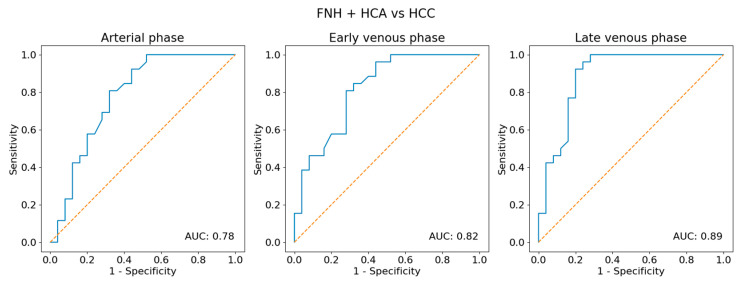
Receiver operator characteristic (ROC) curve analysis for predicting the focal nodular hyperplasia + hepatocellular adenoma (FNH + HCA) group or hepatocellular carcinoma (HCC) group based on the echogenicity of a tumor.

**Figure 10 diagnostics-13-02025-f010:**
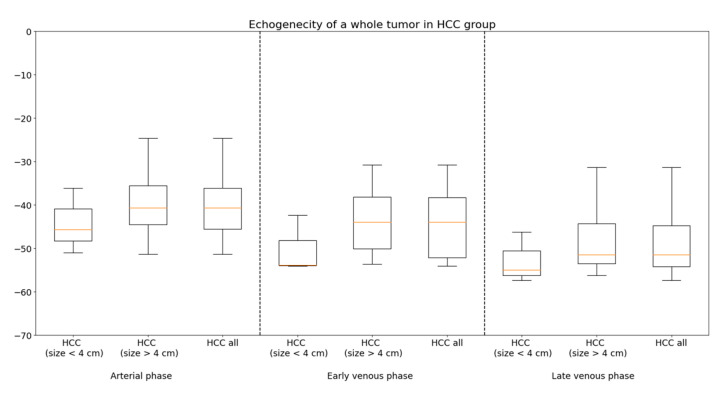
Record of the values of enhancement of the whole tumor (hepatocellular carcinoma (HCC) subgroups based on the size of the lesion) recorded in the arterial, portal venous and late venous phases.

**Figure 11 diagnostics-13-02025-f011:**
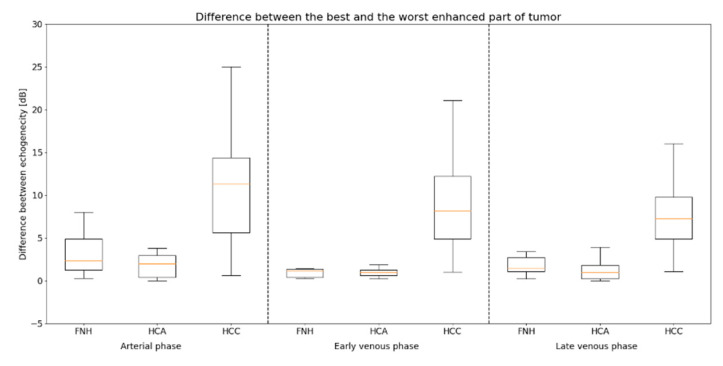
Record of the difference in the tumor enhancement between the best and the worst enhanced part of the tumor. Focal nodular hyperplasia (FNH), hepatocellular adenoma (HCA), hepatocellular carcinoma (HCC).

**Table 1 diagnostics-13-02025-t001:** Performed diagnostics.

Lesion	Sex Male/Female	Computed Tomography (+/N)	Magnetic Resonance (+/N)	Biopsy/Post-Operative Verification (+/N)
HCA	6/8	3/16	16/16	2/16
FNH	0/10	2/10	10/10	0/10
HCC	21/7	28/28	8/28	28/28

**Table 2 diagnostics-13-02025-t002:** Demographic and size of the lesion analysis.

Patient Characteristics		FNH	HCA	*p*-Value (FNH vs. HCA)	FNH + HCA	HCC	*p*-Value (FNH + HCA vs. HCC)
Sex	Male	0	6	-	6	21	0.00012
	Female	11	12		23	7	
Age	Median	32	30	0.84	30	68	0.000123
	Range	(22–41)	(19–80)		(19–80)	(39–85)	
Smaller dimension of a lesion (CEUS) [cm]	Median	2.6	2.15	0.86	2.27	5.12	0.00000556
	Range	(0.9–3.6)	(0.62–7.69)		(0.62–7.69)	(1.59–16.61)	
Larger dimension of a lesion (CEUS) [cm]	Median	3.34	2.88	0.86	3.1	5.82	0.00000033
	Range	(1.41–4.6)	(0.66–8.2)		(0.66–8.2)	(2.88–18.63)	

## Data Availability

The data presented in this study are available on request from the corresponding author.

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
