# Peer review of "Usefulness of Contrast-Enhanced Ultrasound in the Differentiation between Hepatocellular Carcinoma and Benign Liver Lesions"

_diagnostics, 2023, doi:10.3390/diagnostics13122025_

Round 1
Reviewer 1 Report
Dear Authors, thank you very much for the opportunity to review this interesting manuscript.
However, some significant revisions are required to deem it fit for publication in this important journal:
- in the introduction/discussion, some reference to guidelines and protocols for the diagnosis are needed, as well as a note to the role and semiotics of other imaging modalities; i am suggesting a couple, but feel free to add whichever you prefer:
https://pubmed.ncbi.nlm.nih.gov/25376286/
https://pubmed.ncbi.nlm.nih.gov/26312574/;
- the demographics and data about enrolled patients should appear in the result section; M&M is more fit for a description of inclusion/exclusion criteria when talking about the population;
- table 2 would benefit of statistical testing to weight the differences and similarities between groups;
- figurers depicting b mode ultrasound would benefit from a higher resolution/smaller canvas size;
- the analysis and the box plot are sufficient, but a matrix of ROC curve comparing the dB values among lesions, or even just HCC vs the rest, would be optimal
Reviewer 2 Report
Authors present a retrospective study on the role of ceus on the differentiation between HCC and benign liver lesions is required.
The paper is well written although below you can find some suggestions:
Why authors didn't provide accuracy values (spe and sen) for malignant histopathological proved lesions?
Abstract: please do not use abbreviation when using for the first time a word
(ex HCA, FNH etc..)
On table 2 just provide one between medians /range and Mean/sd (suggest to put the one excluded in the supplementary materials)
Round 2
Reviewer 1 Report
All the relevant major revision have been addressed